# Environmental Life Cycle Assessment of Two Types of Flexible Plastic Packaging under a Sustainable Circular Economy Approach

**Börçe Tunçok-Çeşme** [1,*] **, Eren Yıldız-Geyhan** [2] **and Gökçen Alev Çiftçioğlu** [1]

1 Department of Chemical Engineering, Marmara University, İstanbul 34854, Türkiye; gciftcioglu@marmara.edu.tr
2 Maltepe Municipality, İstanbul 34843, Türkiye; rnyildiz@yahoo.com
* Correspondence: borcetuncok@gmail.com

**Abstract:** While it is of great importance to evaluate plastic waste within the framework of a circular economy today, it is also of great importance to evaluate flexible plastic packaging, which is increasingly used in order to prevent environmental problems. To avoid the disadvantages experienced in recycling due to its multilayer nature, in this study, a life cycle assessment was made for flexible packages consisting of PET/metallized PET/PE and PP/metallized PP/PP with the data provided by the R&D Center of Elif Plastik Ambalaj Sanayi ve Tic. A.Ş.-Huhtamaki Flexibles Istanbul. Within the scope of this evaluation, two types of flexible packaging were analyzed, and an optimal flexible packaging structure for Türkiye was revealed by comparing different scenarios based on different methods in SimaPro 8.1.1.16. LCA was performed for both packages with cumulative energy demand (CED) and CML-IA methods. Four scenarios with different amounts of recycled raw materials were compared against the existing system and a fifth scenario, where electricity is assumed to be obtained from solar energy. Overall, we found that the largest environmental impact was in the existing system. However, despite being a renewable energy source, we observed that the solar energy scenario had almost as significant an impact as the existing system. When scenarios involving recycled raw materials were examined, we clearly observed that as the amount of recycled raw materials increased, the environmental impact decreased. Therefore, it emerged that the scenario with the highest amount of recycled raw materials is the most optimal scenario in many respects. There are clear differences in the results due to differences in plastic types. This study, conducted with real data, is highly important for the flexible packaging literature. A table has been provided for changing the type of plastic, changing the source of electricity generation, and reducing waste by using recycled raw materials in order to make flexible packaging more environmentally beneficial.

**Keywords:** circular economy; LCA; life cycle assessment; flexible packaging; environmental life cycle assessment (ELCA); SimaPro; sustainability

## 1. Introduction

After the industrial revolution, in order to meet increasing demand, plastic has a big place in our lives due to its many advantages such as low cost, light weight, and durability. Plastic production, which was 15 million tons in 1964, has increased significantly over the years, reaching 400.3 million tons by 2022, increasing more than 25 times in 58 years [1,2]. Approximately 39% of the plastics produced in Türkiye and also in Europe are used as packaging material [3,4]. One of the packaging types that has increased in use in recent years is flexible packaging.

The main purpose of flexible packaging is storage and packaging of products. Flexible packaging is generally used in the packaging of electronics, medicine, chemicals, food, and beverages. Bags, pouches, sachets, squeezable containers, films, and wraps are the types of flexible packaging. In order to increase the protection and storage properties of

commonly used monolayer packaging materials such as polyethylene teraphtalate (PET), polyethylene (PE), and polypropylene (PP), designing them as multilayer packaging with other materials such as oriented PET, metallized PET, oriented PP (OPP), etc. has enabled the production of packaging with the desired properties [5]. Plastics can be used in flexible packaging, as well as paper and aluminum foil. The most used material is plastic with a ratio of approximately 76.8%. It is followed by paper with 11.4% and then aluminum with 9.8% [6].

Compared to rigid packaging, flexible packaging has a larger market share because it is lighter, has lower production costs, lower transportation costs, and, most importantly, has a longer shelf life. Although 80% of flexible packaging is recyclable because it is a mono-material consisting of PE or PP, flexible packaging has a 14% recycling ratio in the world because of its structure, technical issues, and recovery, collection, and classification problems [6,7].

The global volume of consumer flexible packaging is estimated to have increased from 27.4 million tons in 2017 to 33.5 million tons in 2023 and market value is estimated at USD 291.2 billion in 2023 [8]. In Türkiye, flexible packaging production reached 1.18 million tons and USD 3112 million in 2021 and is expected to increase continuously [9].

According to Circular Economy for Flexible Packaging (CEFLEX) reports, major brands owners guarantee to make 100% of packaging recyclable, reusable, or compostable by 2025. At the same time, European Union countries have reported that they aim to recycle 55% of plastics by 2030. In this regard, the circular economy approach, which aims at waste reduction rather than the linear economy that adopts the "take, make, and dispose" approach, is also of great importance for flexible packaging [7].

Life cycle assessment examines the environmental aspects and possible environmental impacts (such as the use of resources and the environmental consequences of emissions) of a product throughout its life (such as from cradle to grave), from the procurement of raw materials used for its production, to its use, processing at the end of its life, recycling, and final disposal [10,11].

Ögmundarson et al. [12] published an article on the life cycle assessment (LCA) of plastic packaging waste in Iceland. It is of great importance that it is the first for Iceland, and it is also an important publication because it shows that mechanical recycling has more environmental benefits than storing waste in Iceland, even though it includes the effects of exporting waste to different European countries. The study conducted by Foolmaun and Ramheeawon [13] compares the life cycle assessment and social life cycle assessment of PET waste in Mauritius. Four different scenarios were examined to determine the optimal solution for Mauritius. Bassi et al. [14] published a detailed LCA study for PET packaging management strategies in the European Union (EU). With this study, both environmental and social life cycle assessment results focusing on the management of PET waste towards the EU's 2030 targets have been presented. Schmidt and Laner [15] revealed the environmental performance of the management of plastic waste in Germany by an LCA study, examined many impact categories, and presented the factors that Germany should focus on in the long term with a detailed study. Another important study was conducted by Pragati and Maeda [16] for Japan. Environmental impacts of consumer packaging products have been revealed through LCA. It compared plastic products such as PET and high-density polyethylene (HDPE) with products with less environmental impact such as glass bottles, aluminum bottles, paper packaging, and textile bags. It is a cradle-to-grave-focused study and MiLCA software version 2.3 was used. Kan and Miller [17] published an environmental impact article about plastic packaging used in food products. Many impact factors were analyzed, and the results compared. Biona et al. [18] conducted a comparative LCA of plastic and paper packaging bags for the Philippines in 2015. Cradle-to-grave analysis was performed for these two packaging types and many impact factors were examined.

While numerous life cycle assessment (LCA) investigations have been undertaken concerning plastic packaging, there remains a noticeable dearth of research specifically

dedicated to flexible packaging. The existing studies predominantly center around recycling practices within the realm of flexible packaging. Below, we outline the array of studies exclusively concentrated on LCA within this domain.

Ahamed et al. [19] carried out an LCA study and they offered conversion to multi-walled carbon nanotubes as a solution to increase the recycling rate, which is a problem especially for flexible packaging. Comparisons were made for different plastic contents, and LCA was performed to examine many impact factors. Siracusa et al. [20] published an LCA study for multilayer plastic bags. System boundaries were selected from cradle to factory-gate. How much change the amount of recycled PA used in the process caused was examined with many parameters, and its effect on multilayer food packaging was revealed in this study. Horodytska et al. [21] carried out a review about plastic flexible film waste management. They analyzed postindustrial and postconsumer waste management. This study focused on recycling processes and reviewed all LCA studies about this issue. They revealed that there are many LCA deficiencies in this regard. He et al. [22] carried out an LCA study about flexible packaging processes. They compared the solventless and dry film lamination processes in China. With the aim of cleaner production, they analyzed the results of these processes and they searched for effective methods and environmentally and performance-friendly options for food packaging. Farrukh et al. [6] published a systematic review on environmental issues of flexible packaging. This study indicated that the number of articles about flexible packaging has increased year by year. As expected, it was revealed that most of these articles belonged to European Union countries. At the same time, it clearly stated which methods were used, what outputs they gave, and what solutions they offered for the articles that have relevant keywords. Kruefer et al. [23] published an LCA study that was conducted using different methods and impact factors for two different-sized HDPE bottles and multilayer pouches. The study is a regional study focusing on Europe, Latin America, and Australia. Costamagna et al. [24] conducted an LCA study to find a solution to recycling, which is one of the most important problems of multilayer flexible packaging. By dissolving with monoethylene glycol, various impact factors were examined through the life cycle evaluation of polyethylene-polyamide films and an optimal solution for recycling processes was presented.

In this study, scenarios were created assuming that changes would be made in the production stages of two flexible packages, and the aim was to find the optimal sustainable production scenario with the least environmental impact by comparing the scenarios with life cycle assessment.

## 2. Materials and Methods

Life cycle assessment methodology was used to compare the scenarios produced for two different flexible packaging types which are PET/metallized PET/PE and PP/metallized PP/PP. According to ISO14040 and ISO14044, life cycle assessment (LCA) has four main steps. These are goal and scope definition, life cycle inventory, life cycle analysis, and interpretation of the results [10,11]. Also, SimaPro 8.1.1.16 (PhD) was used as LCA software for modeling.

### 2.1. Goal and Scope Definition

The main purpose of this study is to compare the PET/metallized PET/PE flexible packaging with the PP/metallized PP/PP flexible packaging and to find the optimal scenario among different scenarios within the framework of circular economy.

System boundaries for flexible packaging are given in Figure 1 and Figure 2. These system boundaries were created based on the process flows in Huhtamaki Flexibles Türkiye. To find the optimal sustainable scenario, which is one of the aims of the study, various scenarios were produced by changing the quantity of recycled raw materials and energy source within the system boundaries. The functional unit was chosen as 1 lot of packaging for both packaging types.

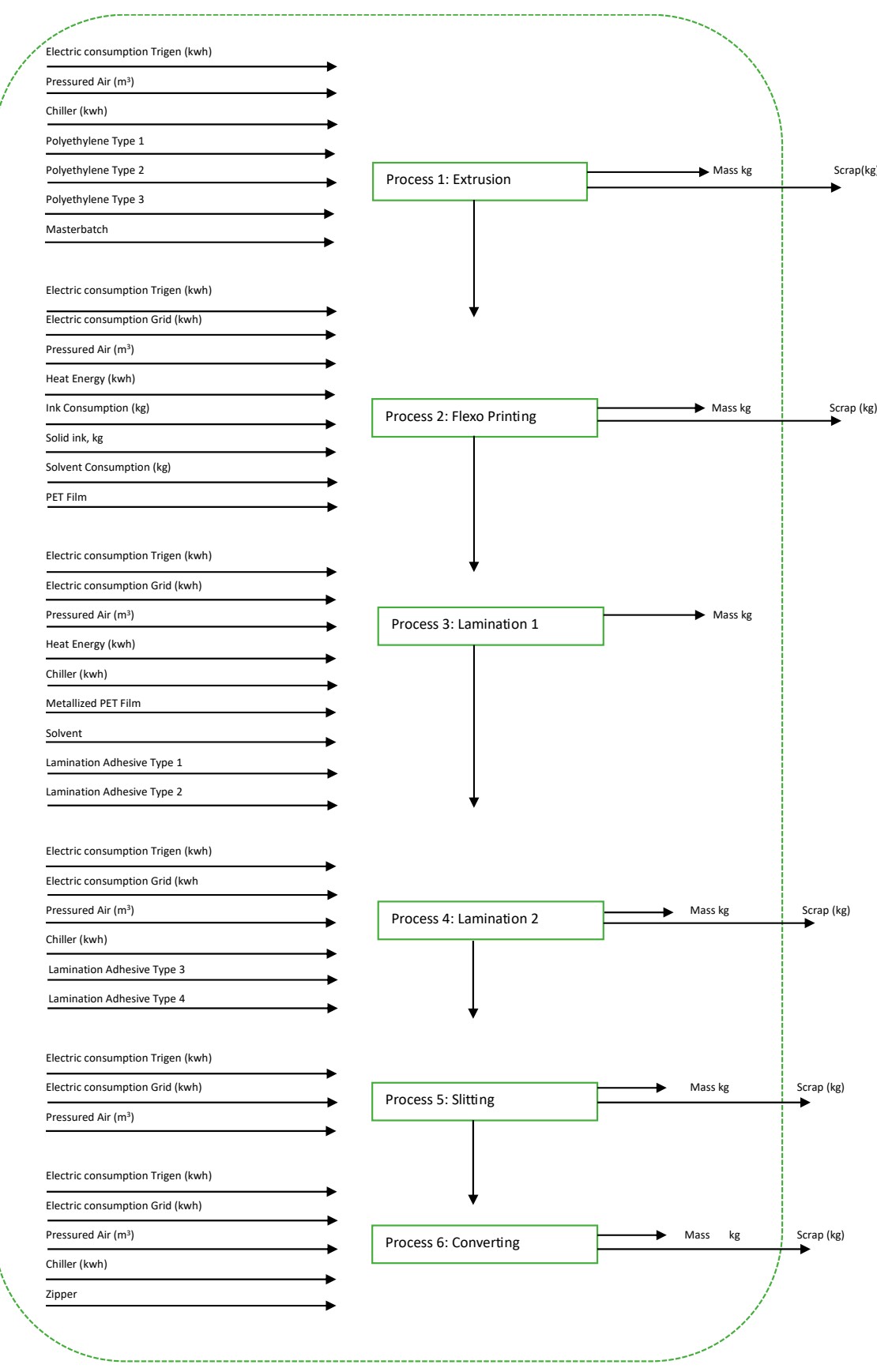

**Figure 1.** System boundaries of PET/Metallized PET/PE.

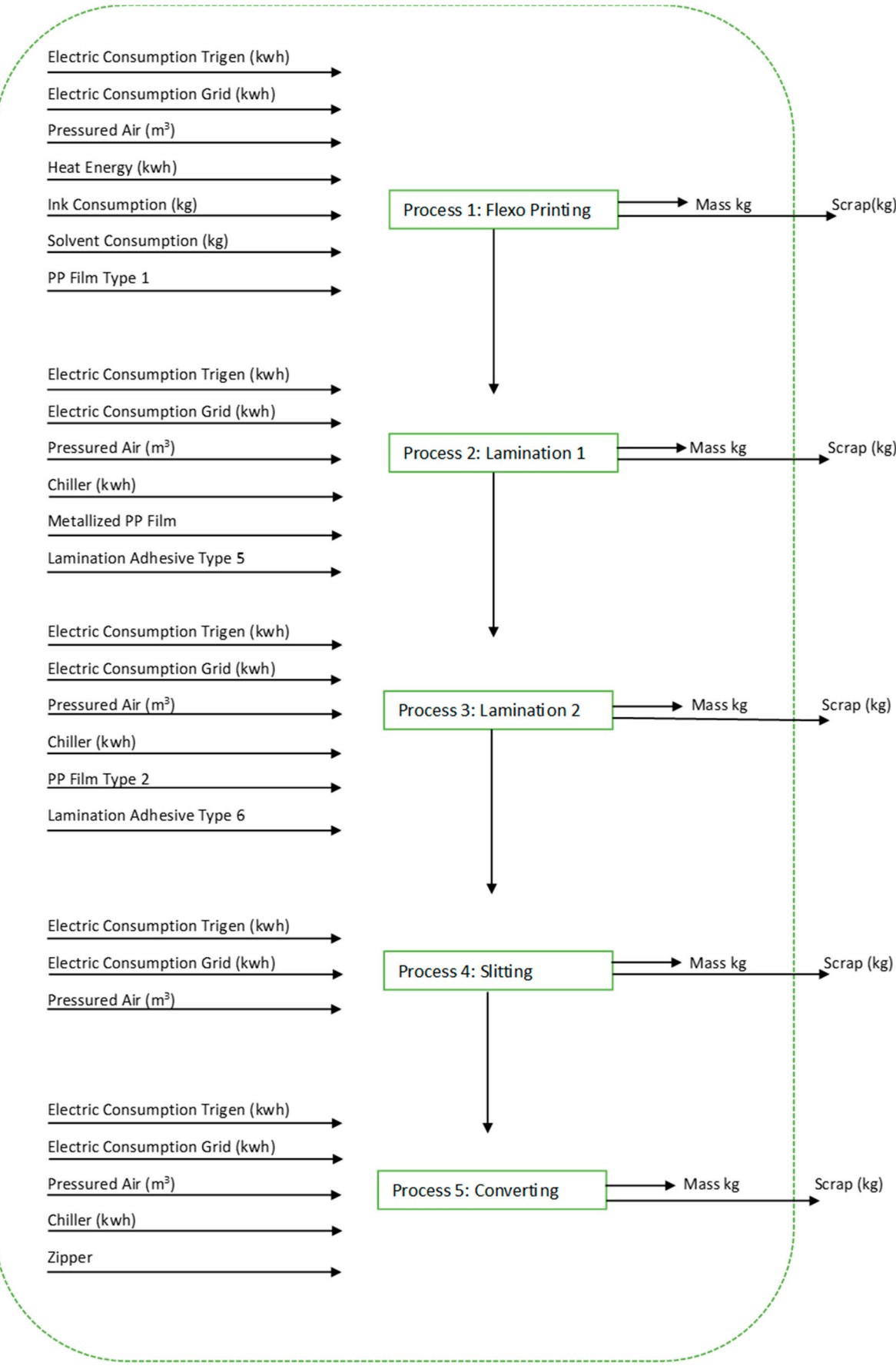

**Figure 2.** System boundaries of PP/Metallized PP/PP.

### 2.2. Inventory Data Collection

In this study, data of two types of flexible packaging produced by Huhtamaki Flexibles Türkiye were used. All energy values, raw materials, and input–outputs used when producing these packages are based on actual data. In addition to all this data, medium-voltage electricity was used during the production phase. The density of the air used is 1.225 kg/m³. For metallized PET and metallized PP, the aluminum content was neglected because it was in nano size. In the scenarios, the impact of using recycled materials at different stages of production and increasing their ratios on the Existing System has been observed. The used data are provided in Appendix A.

#### 2.2.1. Scenario Details of PET/Metallized PET/PE

Existing System (ES): Existing System describes basic flexible packaging production data with specific assumptions taken from the actual system of the R&D Center of Elif Plastik Ambalaj Sanayi ve Tic. A.Ş.-Huhtamaki Flexibles Istanbul. In this scenario, in order to protect intellectual property, the Existing System is extrapolated from the operational framework by excluding mechanical recycling of PE films.

Scenario 1 (S1): In the Extrusion stage, it is assumed that 10% of the input mass of PE Type 2 and PE Type 1 is recycled PE Type 4 and PE Type 3, respectively. Similarly, in the Printing stage, it is assumed that 10% of the input mass of PET is recycled PET. Therefore, in Scenario 1, it is assumed that 90% of the input masses of PE Type 2, PE Type 1, and PET are virgin, while 10% are recycled when entering the system.

Scenario 2 (S2): In the Extrusion stage, it is assumed that 25% of the input mass of PE Type 2 and PE Type 1 is recycled PE Type 2 and PE Type 1, respectively. Similarly, in the Printing stage, it is assumed that 25% of the input mass of PET is recycled PET. Therefore, in Scenario 1, it is assumed that 75% of the input masses of PE Type 2, PE Type 1, and PET are virgin, while 25% are recycled when entering the system.

Scenario 3 (S3): In the Extrusion stage, it is assumed that 50% of the input mass of PE Type 2 and PE Type 1 is recycled PE Type 2 and PE Type 1, respectively. Similarly, in the Printing stage, it is assumed that 50% of the input mass of PET is recycled PET. Therefore, in Scenario 1, it is assumed that 50% of the input masses of PE Type 2, PE Type 1, and PET are virgin, while 10% are recycled when entering the system.

Scenario 4 (S4): In the Extrusion stage, it is assumed that 75% of the input mass of PE Type 2 and PE Type 1 is recycled PE Type 2 and PE Type 1, respectively. Similarly, in the Printing stage, it is assumed that 75% of the input mass of PET is recycled PET. Therefore, in Scenario 1, it is assumed that 25% of the input masses of PE Type 2, PE Type 1, and PET are virgin, while 10% are recycled when entering the system.

Scenario 5 (S5): It is assumed that all used electricity is generated from solar energy without any changes to the Existing System.

The details of all stages of production for PET/Metallized PET/PE are provided in Table 1 on a scenario basis.

**Table 1.** Table of alternative scenarios for PET/metallized PET/PE.

| PET/Metallized PET/PE | | | | | | |
|---|---|---|---|---|---|---|
| **Process** | **ES** | **S1** | **S2** | **S3** | **S4** | **S5** |
| Extrusion | No recycling with usual electric consumption | 10% recycled PE Type 2, PE Type 1 + 90% virgin PE Type 2, PE Type 1 | 25% recycled PE Type 2, PE Type 1 + 75% virgin PE Type 2, PE Type 1 | 50% recycled PE Type 2, PE Type 1 + 50% virgin PE Type 2, PE Type 1 | 75% recycled PE Type 2, PE Type 1 + 25% virgin PE Type 2, PE Type 1 | Electric consumption comes from photovoltaic energy |

| | | | PET/Metallized PET/PE | | | |
|---|---|---|---|---|---|---|
| **Process** | **ES** | **S1** | **S2** | **S3** | **S4** | **S5** |
| Printing | No recycling with usual electric consumption | 10% recycled PET + 90% virgin PET | 25% recycled PET + 75% virgin PET | 50% recycled PET + 50% virgin PET | 75% recycled PET + 25% virgin PET | Electric consumption comes from photovoltaic energy |
| Lamination 1 | No recycling with usual electric consumption | No recycling with usual electric consumption | No recycling with usual electric consumption | No recycling with usual electric consumption | No recycling with usual electric consumption | Electric consumption comes from photovoltaic energy |
| Lamination 2 | No recycling with usual electric consumption | No recycling with usual electric consumption | No recycling with usual electric consumption | No recycling with usual electric consumption | No recycling with usual electric consumption | Electric consumption comes from photovoltaic energy |
| Slitting | No recycling with usual electric consumption | No recycling with usual electric consumption | No recycling with usual electric consumption | No recycling with usual electric consumption | No recycling with usual electric consumption | Electric consumption comes from photovoltaic energy |
| Converting | No recycling with usual electric consumption | No recycling with usual electric consumption | No recycling with usual electric consumption | No recycling with usual electric consumption | No recycling with usual electric consumption | Electric consumption comes from photovoltaic energy |

2.2.2. Scenario Details of PP/Metallized PP/PP

Existing System (ES): Existing System describes basic flexible packaging production data with specific assumptions taken from the actual system of the R&D Center of Elif Plastik Ambalaj Sanayi ve Tic. A.Ş.-Huhtamaki Flexibles Istanbul. In this scenario, in order to protect intellectual property, the Existing System is extrapolated from the operational framework by excluding mechanical recycling of PE films.

Scenario 1 (S1): In the Printing stage, it is assumed that 10% of the input mass of PP Type 1 is recycled PP Type 1. Similarly, in the Lamination 2 stage, it is assumed that 10% of the input mass of PP Type 2 is recycled PP Type 2. Therefore, in Scenario 1, it is assumed that 90% of the input masses of PP Type 1 and PP Type 2 are virgin, while 10% are recycled when entering the system.

Scenario 2 (S2): In the Printing stage, it is assumed that 25% of the input mass of PP Type 1 is recycled PP Type 1. Similarly, in the Lamination 2 stage, it is assumed that 25% of the input mass of PP Type 2 is recycled PP Type 2. Therefore, in Scenario 1, it is assumed that 75% of the input masses of PP Type 1 and PP Type 2 are virgin, while 25% are recycled when entering the system.

Scenario 3 (S3): In the Printing stage, it is assumed that 50% of the input mass of PP Type 1 is recycled PP Type 1. Similarly, in the Lamination 2 stage, it is assumed that 50% of the input mass of PP Type 2 is recycled PP Type 2. Therefore, in Scenario 1, it is assumed that 50% of the input masses of PP Type 1 and PP Type 2 are virgin, while 50% are recycled when entering the system.

Scenario 4 (S4): In the Printing stage, it is assumed that 75% of the input mass of PP Type 1 is recycled PP Type 1. Similarly, in the Lamination 2 stage, it is assumed that 75% of the input mass of PP Type 2 is recycled PP Type 2. Therefore, in Scenario 1, it is assumed

that 25% of the input masses of PP Type 1 and PP Type 2 are virgin, while 75% are recycled when entering the system.

Scenario 5 (S5): It is assumed that all used electricity is generated from solar energy without any changes to the Existing System.

The details of all stages of production for PP/Metallized PP/PP are provided in Table 2 on a scenario basis.

**Table 2.** Table of alternative scenarios for PP/Metallized PP/PP.

| | PP/Metallized PP/PP | | | | | |
|---|---|---|---|---|---|---|
| Process | ES | S1 | S2 | S3 | S4 | S5 |
| Printing | No recycling with usual electric consumption | 10% recycled PP Type 1 + 90% virgin PP Type 1 | 25% recycled PP Type 1 + 75% virgin PP Type 1 | 50% recycled PP Type 1 + 50% virgin PP Type 1 | 75% recycled PP Type 1 + 25% virgin PP Type 1 | Electric consumption comes from photovoltaic energy |
| Lamination 1 | No recycling with usual electric consumption | No recycling with usual electric consumption | No recycling with usual electric consumption | No recycling with usual electric consumption | No recycling with usual electric consumption | Electric consumption comes from photovoltaic energy |
| Lamination 2 | No recycling with usual electric consumption | 10% recycled PP Type 2 + 90% virgin PP Type 2 | 25% recycled PP Type 2 + 75% virgin PP Type 2 | 50% recycled PP Type 2 + 50% virgin PP Type 2 | 75% recycled PP Type 2 + 25% virgin PP Type 2 | Electric consumption comes from photovoltaic energy |
| Slitting | No recycling with usual electric consumption | No recycling with usual electric consumption | No recycling with usual electric consumption | No recycling with usual electric consumption | No recycling with usual electric consumption | Electric consumption comes from photovoltaic energy |
| Converting | No recycling with usual electric consumption | No recycling with usual electric consumption | No recycling with usual electric consumption | No recycling with usual electric consumption | No recycling with usual electric consumption | Electric consumption comes from photovoltaic energy |

### 2.3. Life Cycle Impact Assessment

This is the third step of life cycle assessment. Cumulative energy demand and CML-IA (Center of Environmental Science of Leiden University), which are among the most used methods, were used for the environmental impact analysis of the system. In the CED analysis, all categories of non-renewable fossil, non-renewable nuclear, non-renewable biomass, renewable biomass, renewable wind–solar–geothermal, and renewable water were taken into consideration. In the CML-IA analysis, 11 impact categories were examined, which are abiotic depletion, abiotic depletion (fossil), global warming (GWP100a), ozone layer depletion (ODP), human toxicity, freshwater aquatic ecotoxicity, marine aquatic ecotoxicity, terrestrial ecotoxicity, photochemical oxidation, acidification, and eutrophication.

### 2.4. Interpretation

In the last stage, the interpretation step, all results are analyzed, sensitivity is checked, and limitations are given along with the results. At the same time, recommendations are given in line with the information given at the stage of the goal/scope of the study [25].

## 3. Results

### *3.1. Environmental Impact Analysis (EIA)*

3.1.1. PET/Metallized PET/PE

This type of flexible packaging has been examined by two analysis methods. These are the most used methods: cumulative energy demand (CED) and CML-IA.

Cumulative Energy Demand (CED)

Cumulative energy demand (CED) is a method used to evaluate the energy consumed throughout the system boundaries, assessing the energy consumed from cradle to gate for this flexible packaging. When examining energy types and scenarios, the most significant change is observed in the non-renewable nuclear energy type. In Scenario 5, assuming the use of the Existing System along with solar energy and non-renewable sources (fossil, nuclear, and biomass), similar results are observed. This result indicates that the difference is due to the energy consumed during the production of solar panels, which results in lower energy consumption during their use.

In scenarios assuming the use of recycled materials for sustainability, it is observed that energy consumption decreases as the use of recycled materials increases. Looking at renewable sources, as expected, the lowest energy consumption is seen in the wind, solar, and geothermal categories in Scenario 5, assuming the use of solar energy. When examining water-based renewable energy consumption, although a decrease is observed compared to the Existing System in the scenario where solar energy is used, it is seen that consumption decreases as the use of recycled materials increases. The same applies to the renewable biomass energy type as given in Figure 3.

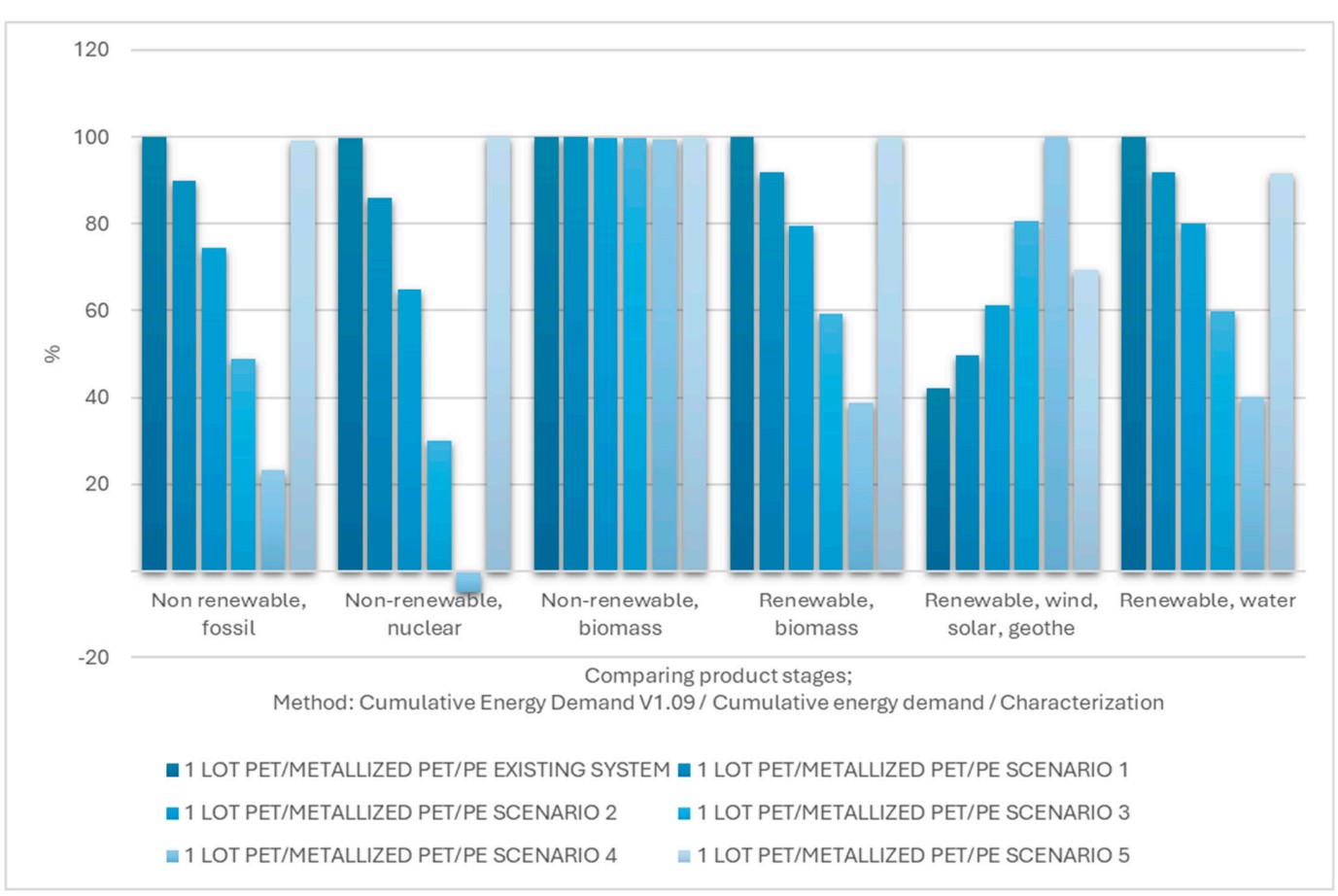

**Figure 3.** Cumulative energy demand characterization analysis results of 5 scenarios for PET/metallized PET/PE.

When the total energy amount is examined using single-score analysis regardless of the energy type as seen in Figure 4, the least change is observed in Scenario 5, where solar energy is assumed to be used with the Existing System. With an approximate change of 0.61%, we can say that it has the closest energy consumption to the Existing System. In scenarios with added recycled materials, we also observe that the change rate increases as the amount of recycled materials increases. When we add recycled materials up to 10% of the total mass, we achieve a change of 10.26%, when we add up to 25%, we achieve a change of 25.67%, when we add up to 50%, we achieve a change of 51.33%, and when we add up to 75%, we achieve a change of 76.91%.

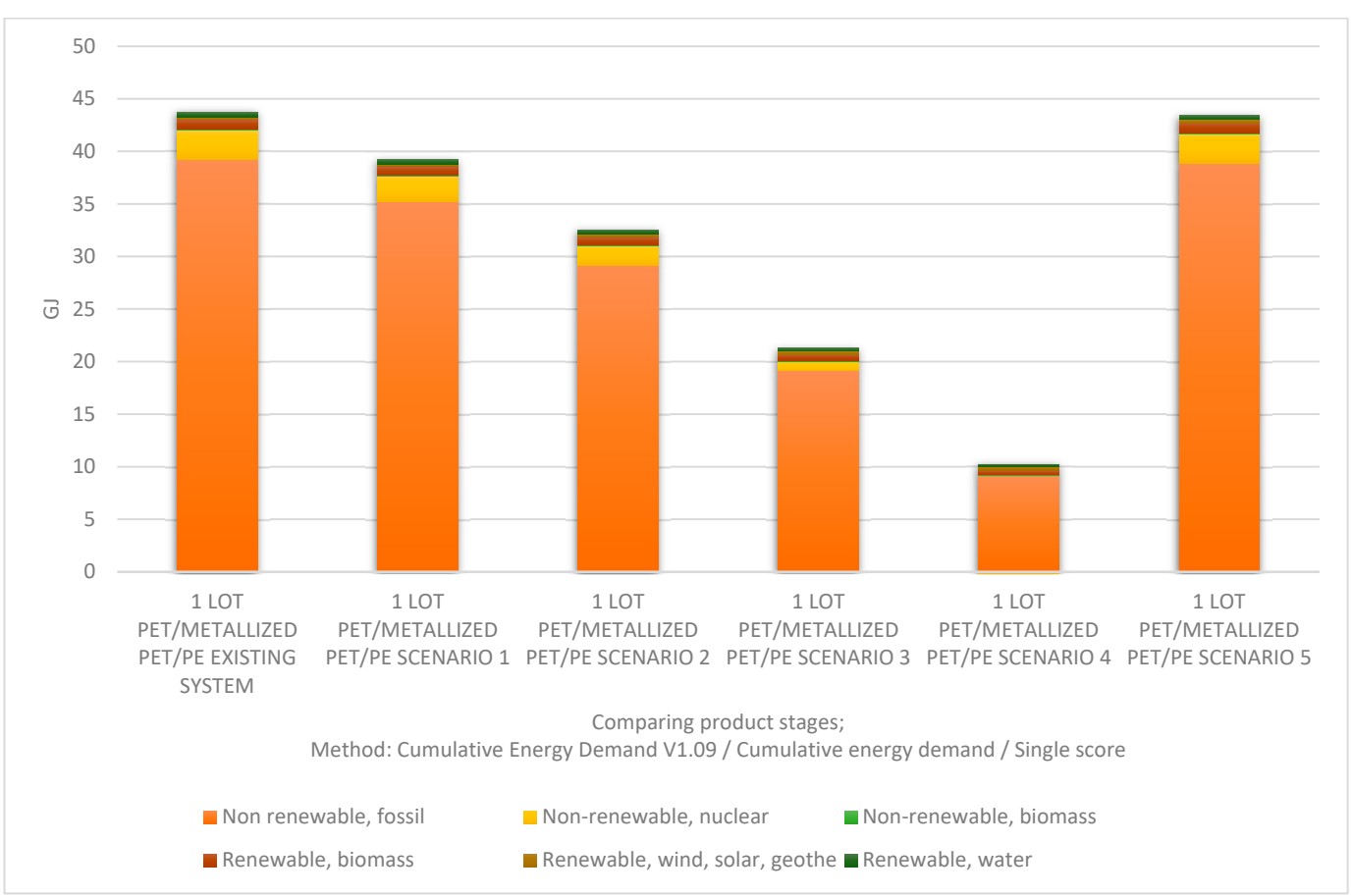

**Figure 4.** Cumulative energy demand single score analysis results of 5 scenarios for PET/metallized PET/PE.

CML-IA

Using SimaPro software version 8.1.1.16(PhD), the Existing System and five different scenarios were analyzed with the CML-IA method for one lot of PET/metallized PET/PE flexible packaging. All results of this analysis are given in Figure 5 and Table 3. In CML-IA, the effect of the process is presented for eleven categories which are shown in the titles of Figure 5.

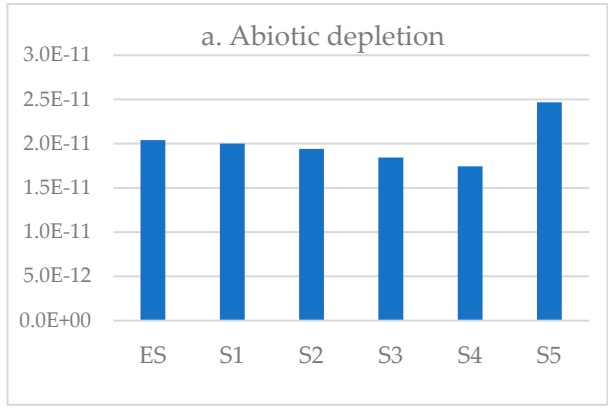

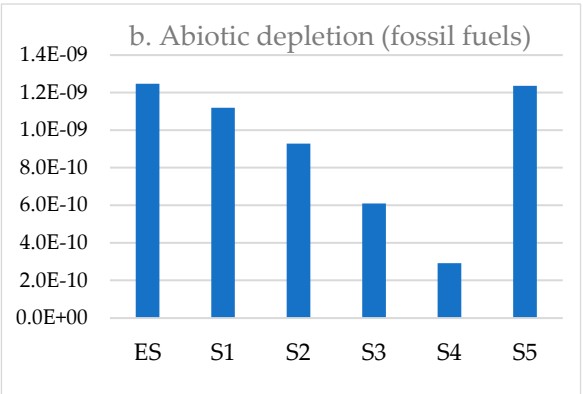

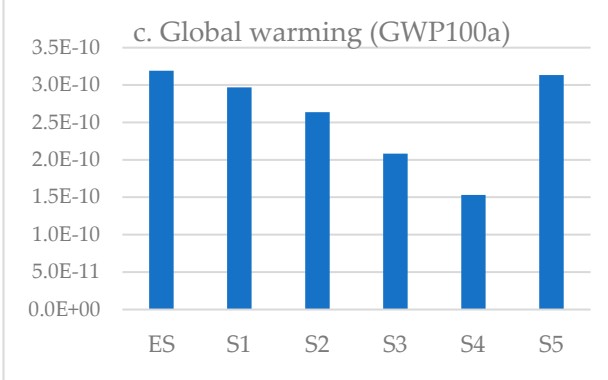

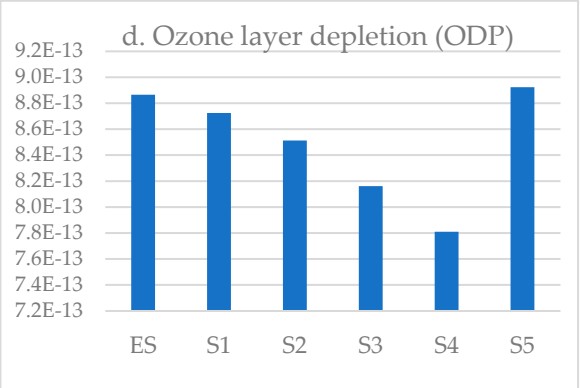

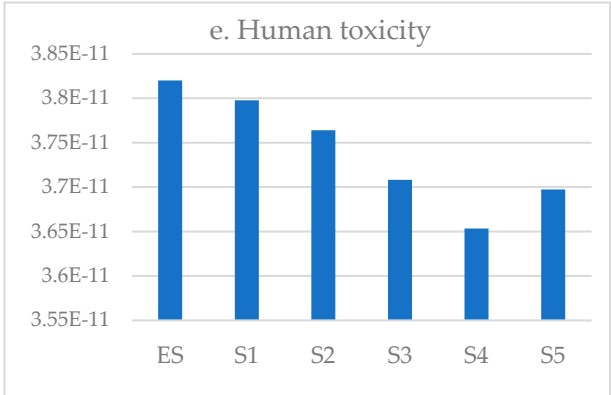

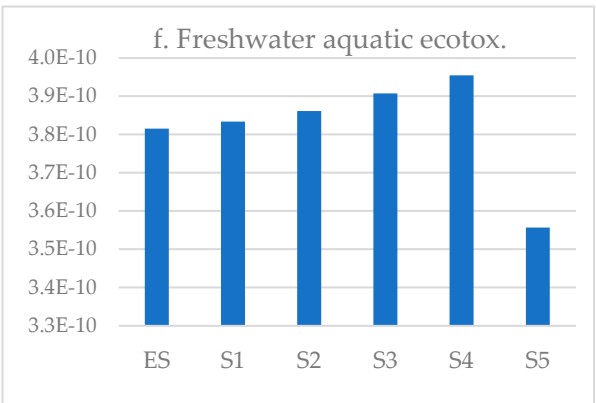

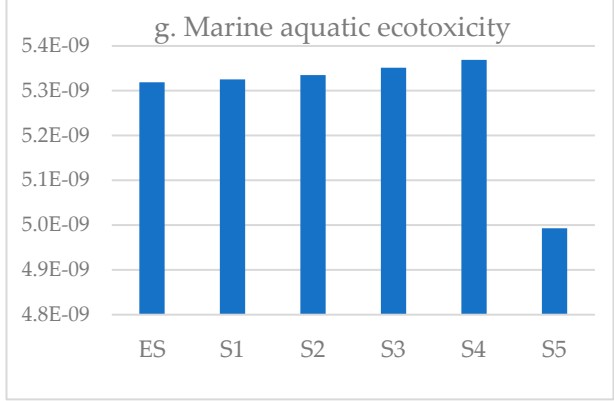

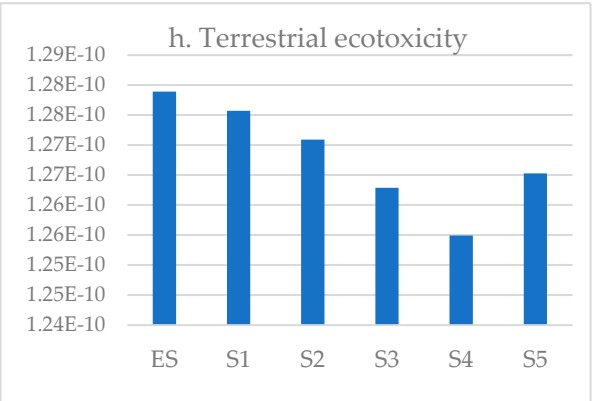

**Figure 5.** *Cont.*

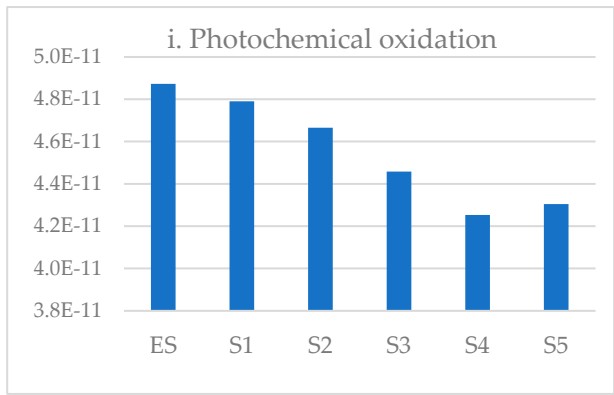

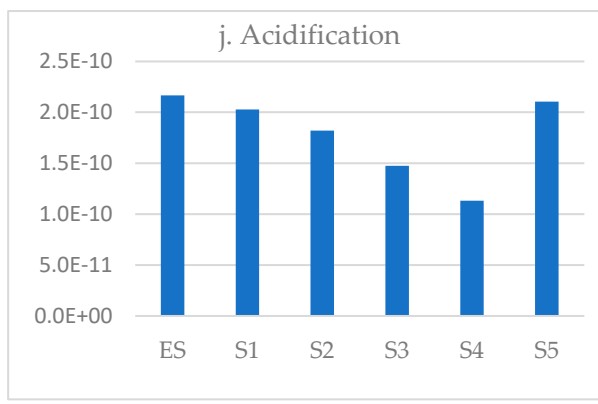

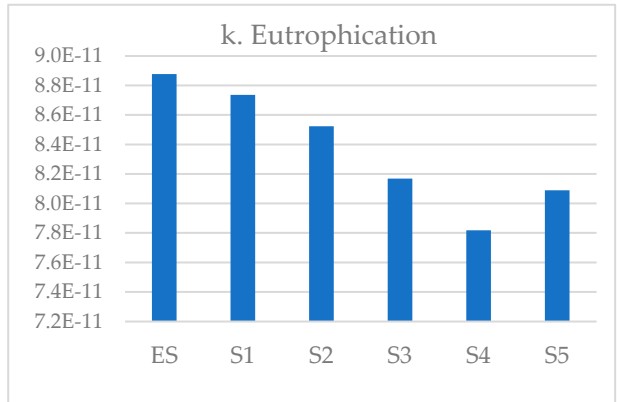

**Figure 5.** Comparative analysis of alternative scenarios and Existing System in terms of (**a**) abiotic depletion, (**b**) abiotic depletion (fossil), (**c**) global warming (GWP100a), (**d**) ozone layer depletion (ODP), (**e**) human toxicity, (**f**) freshwater aquatic ecotoxicity, (**g**) marine aquatic ecotoxicity, (**h**) terrestrial ecotoxicity, (**i**) photochemical oxidation, (**j**) acidification, (**k**) eutrophication.

**Table 3.** Impact assessment results of Existing System and alternative scenarios with CML-IA method.

| | Abiotic Depletion | Abiotic Depletion (Fossil Fuels) | Global Warming (GWP100a) | Ozone Layer Depletion (ODP) | Human Toxicity | Freshwater Aquatic Ecotoxicity | Marine Aquatic Ecotoxicity | Terrestrial Ecotoxicity | Photochemical Oxidation | Acidification | Eutrophication |
|---|---|---|---|---|---|---|---|---|---|---|---|
| ES | $2.04 \times 10^{-11}$ | $1.25 \times 10^{-9}$ | $3.19 \times 10^{-10}$ | $8.87 \times 10^{-13}$ | $3.82 \times 10^{-11}$ | $3.82 \times 10^{-10}$ | $5.32 \times 10^{-9}$ | $1.28 \times 10^{-10}$ | $4.87 \times 10^{-11}$ | $2.17 \times 10^{-10}$ | $8.88 \times 10^{-11}$ |
| S1 | $2.0 \times 10^{-11}$ | $1.12 \times 10^{-9}$ | $2.97 \times 10^{-10}$ | $8.72 \times 10^{-13}$ | $3.8 \times 10^{-11}$ | $3.83 \times 10^{-10}$ | $5.33 \times 10^{-9}$ | $1.28 \times 10^{-10}$ | $4.79 \times 10^{-11}$ | $2.03 \times 10^{-10}$ | $8.74 \times 10^{-11}$ |
| S2 | $1.94 \times 10^{-11}$ | $9.28 \times 10^{-10}$ | $2.64 \times 10^{-10}$ | $8.51 \times 10^{-13}$ | $3.76 \times 10^{-11}$ | $3.86 \times 10^{-10}$ | $5.33 \times 10^{-9}$ | $1.27 \times 10^{-10}$ | $4.67 \times 10^{-11}$ | $1.82 \times 10^{-10}$ | $8.52 \times 10^{-11}$ |
| S3 | $1.84 \times 10^{-11}$ | $6.09 \times 10^{-10}$ | $2.08 \times 10^{-10}$ | $8.16 \times 10^{-13}$ | $3.71 \times 10^{-11}$ | $3.91 \times 10^{-10}$ | $5.35 \times 10^{-9}$ | $1.26 \times 10^{-10}$ | $4.46 \times 10^{-11}$ | $1.47 \times 10^{-10}$ | $8.17 \times 10^{-11}$ |
| S4 | $1.74 \times 10^{-11}$ | $2.91 \times 10^{-10}$ | $1.53 \times 10^{-10}$ | $7.81 \times 10^{-13}$ | $3.65 \times 10^{-11}$ | $3.95 \times 10^{-10}$ | $5.37 \times 10^{-9}$ | $1.25 \times 10^{-10}$ | $4.25 \times 10^{-11}$ | $1.13 \times 10^{-10}$ | $7.82 \times 10^{-11}$ |
| S5 | $2.47 \times 10^{-11}$ | $1.24 \times 10^{-9}$ | $3.13 \times 10^{-10}$ | $8.92 \times 10^{-13}$ | $3.7 \times 10^{-11}$ | $3.56 \times 10^{-10}$ | $4.99 \times 10^{-9}$ | $1.272 \times 10^{-10}$ | $4.3 \times 10^{-11}$ | $2.1 \times 10^{-10}$ | $8.09 \times 10^{-11}$ |

- Abiotic depletion

  Abiotic depletion is an impact factor that allows us to assess the potential effects of depletion of non-living resources such as minerals. When examined according to the scenarios, we observe the highest impact in S5, where we assume the use of solar energy. While the Existing System has the second-highest value, we also see that the use of recycled materials, albeit to a small extent, reduces the impact as seen in Figure 5a.

- Abiotic depletion (fossil)

  Abiotic depletion is an impact factor that allows us to assess the potential effects of depletion of non-living resources such as fossil fuels. When examined according to the scenarios, we observe the highest values in the Existing System and in S5, where we assume the use of solar energy. However, since this high impact is not observed during the use of solar energy, it is believed that this effect stems from the production stage of solar panels. Nevertheless, we also observe that the impact decreases as the use of recycled materials increases as given in Figure 5b.

- Global warming (GWP100a)

Global warming potential (GWP) is an impact factor used to assess the contribution of a substance to global warming over a period of 100 years. The GWP of a substance is calculated based on its ability to trap heat in the atmosphere. This allows us to take actions to reduce the environmental impact related to climate change. In Scenario 5, which we assume utilizes solar energy within the Existing System, we observe a similar impact on global warming over 100 years. However, we also notice that as the use of recycled materials increases in scenarios where recycled materials are assumed to be used, the impact decreases as given in Figure 5c.

- Ozone layer depletion (ODP)

In ozone layer depletion, we observe similar outcomes as with global warming. In Scenario 5, where we assume solar energy is used within the existing system, we see comparable results. It is known that the impact of solar energy usage on ozone layer depletion is minimal. We anticipate that this impact stems from the production of solar panels. However, in scenarios where recycled materials are used, we observe a decrease in the impact as the use of recycled materials increases as given in Figure 5d.

- Human toxicity

Human toxicity is an impact factor that allows us to assess the potential harm of a product to human health throughout its life cycle. When comparing Scenario 5, where we assume the use of solar energy within the existing system, we clearly see that the use of solar energy significantly reduces the impact. Additionally, the use of recycled materials also greatly reduces the impact as seen in Figure 5e.

- Freshwater aquatic ecotoxicity

Freshwater aquatic ecotoxicity is an impact factor through which we can assess the adverse effects on freshwater ecosystems. When examining the results provided in Figure 5f, we observe different outcomes compared to previous findings. Firstly, in Scenario 5 where we assume the use of solar energy, we see the most significant decrease compared to the Existing System. Additionally, when we look at scenarios where recycled materials are used, we notice that as the use of recycled materials increases, the impact also increases.

- Marine aquatic ecotoxicity

Marine aquatic ecotoxicity is an impact factor through which we can assess the adverse effects on marine water ecosystems. When examining the results provided in Figure 5g, we observe different outcomes compared to previous findings, yet we encounter similar results to freshwater aquatic ecotoxicity. Firstly, in Scenario 5 where we assume the use of solar energy, we see the most significant decrease compared to the Existing System. Additionally, when we look at scenarios where recycled materials are used, we notice that as the use of recycled materials increases, the impact also increases.

- Terrestrial ecotoxicity

Terrestrial ecotoxicity is an impact factor that enables us to observe the adverse effects on terrestrial ecosystems. Upon examining Figure 5h, we notice that the highest impact is observed in the Existing System. Additionally, as expected, it is observed that the impact decreases as the use of recycled materials increases. In Scenario 5, where we assume the use of solar energy, a decrease in impact compared to the Existing System is also observed.

- Photochemical oxidation

Photochemical oxidation is a chemical reaction type where chemical compounds undergo oxidation or breakdown using light energy. These reactions are triggered by sunlight or other light sources and play a significant role in the formation of air pollution, atmospheric chemical transformations of organic compounds, and various industrial applications. It is an impact factor that emerges as a component of air pollution. Upon examining

this impact factor, it has been observed that the highest impact occurs in the Existing System. Furthermore, it has been observed that the impact decreases when recycled materials and solar energy are used. Particularly, a noticeable decrease in impact has been observed when solar energy is utilized as seen in Figure 5i.

- Acidification

Acidification is an impact factor that refers to the process of increasing environmental acidity of a substance. It encompasses all environmental mediums such as soil, water, and air. Here, we observe the highest impact in the Existing System and in Scenario 5 where we assume the use of solar energy. Furthermore, we believe that the impact in the solar energy system occurs not during its usage but rather during the production of solar panels. In scenarios where recycled materials are used, it has been observed that as the use of recycled materials increases, the impact decreases as given in Figure 5j.

- Eutrophication

Eutrophication is an impact factor through which we observe the adverse effects resulting from excessive nutrient enrichment in aquatic environments. The highest impact is observed in the Existing System. However, the use of recycled materials or solar energy significantly reduces this impact as seen in Figure 5k.

### 3.1.2. PP/Metallized PP/PP

This type of flexible packaging has been examined by two analysis methods. These are the most commonly used methods: cumulative energy demand (CED) and CML-IA.

#### Cumulative Energy Demand (CED)

In the cumulative energy demand analysis, since all indirect and direct energy is analyzed, significantly different results were obtained from PET/metallized PET/PE packaging. Unlike PET/metallized PET/PE packaging, the percentage changes for each energy type are greater in this type of packaging. This is because the plastic type is different. When examined in detail, as given in Figure 6, it is observed that with the increase in the proportion of recycled materials in the first four scenarios, there is not only a change in the non-renewable biomass type but also a linear decrease in non-renewable fossil, non-renewable nuclear, renewable biomass, renewable wind, and renewable water types. In the 5th scenario, where it is assumed that solar energy is used, significant percentage changes are observed. Demand, especially in renewable wind, solar, and geothermal energy types, increases significantly.

When looking at the total energy demand using the single-score method, we observe the highest energy requirement in the Existing System. In second place, with a 1.5% change, we encounter Scenario 5, where it is assumed that solar energy is used. In Scenario 1, where it is assumed that 10% recycled material is used in the input mass, a decrease of 5.89% is achieved, in Scenario 2 where 25% recycled material is assumed to be used, a decrease of 14.61% is observed, in Scenario 3 where 50% recycled material is assumed to be used, a decrease of 29.16% is observed, and in Scenario 4 where 75% recycled material usage is assumed, a decrease of 43.72% is achieved as given in Figure 7.

#### CML-IA

As for one lot of PET/metallized PET/PE flexible packaging, an analysis was made for one lot of PP/metallized PP/PP flexible packaging with the CML-IA method using SimaPro software for the Existing System and five different scenarios. All results are given in Figure 8 and Table 4.

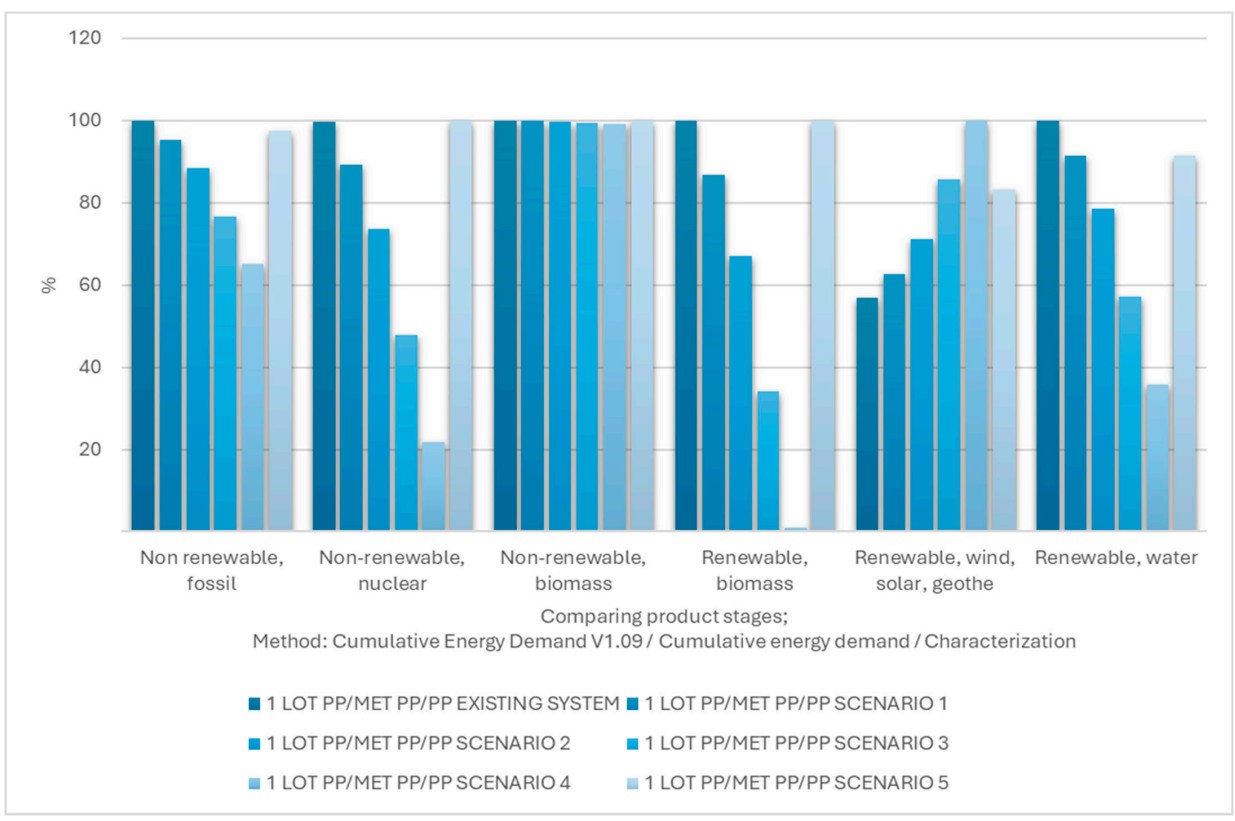

**Figure 6.** Cumulative energy demand characterization analysis results of 5 scenarios for PP/metallized PP/PP.

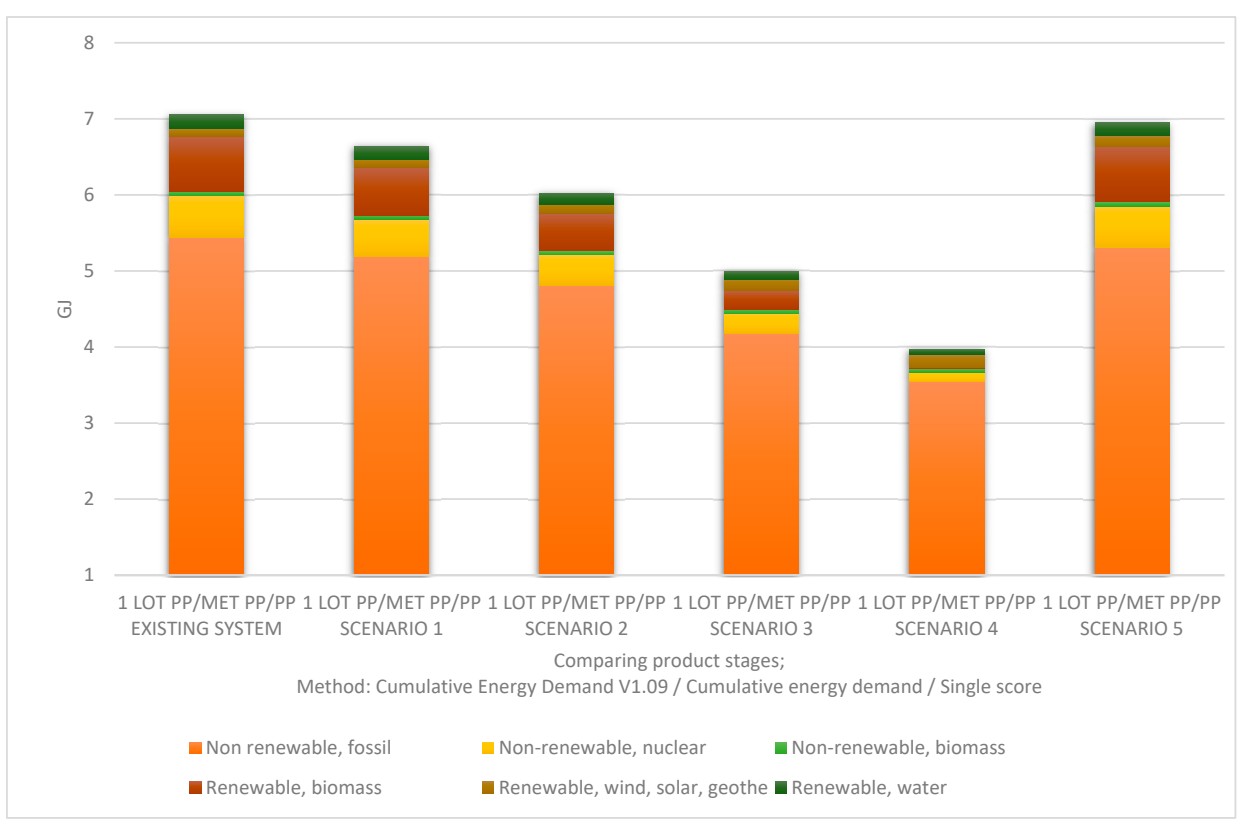

**Figure 7.** Cumulative energy demand single-score analysis results of 5 scenarios for PP/metallized PP/PP.

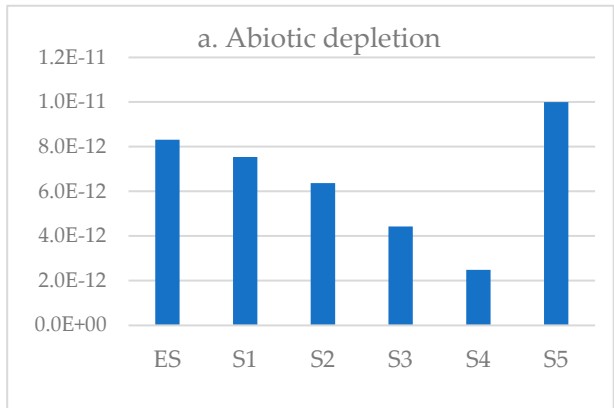

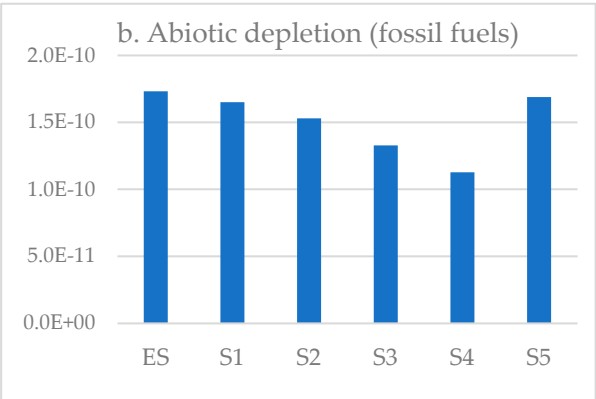

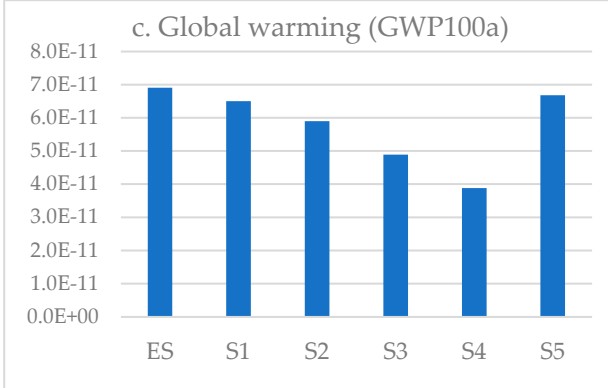

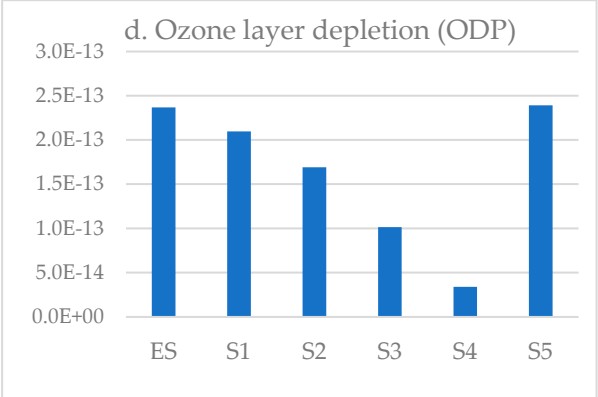

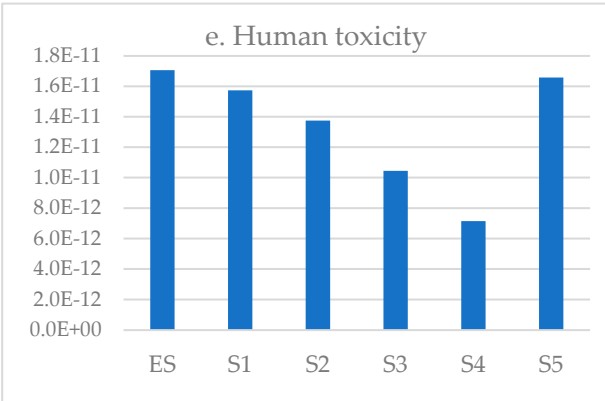

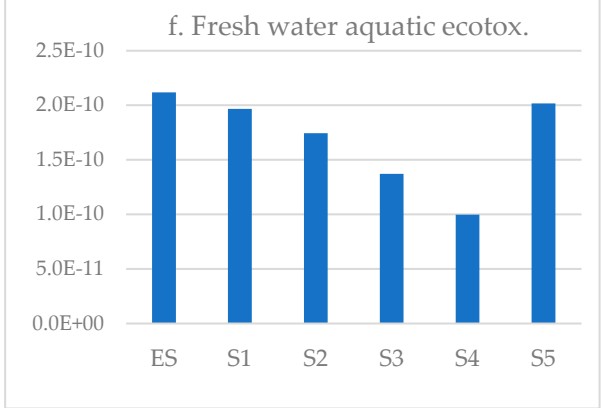

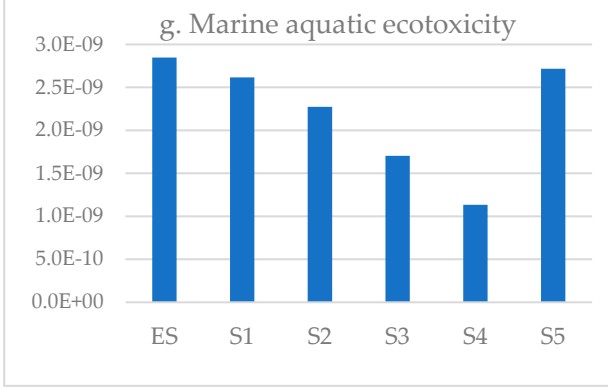



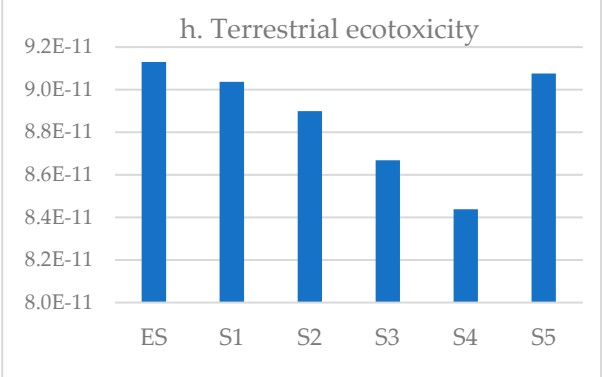

**Figure 8.** *Cont.*

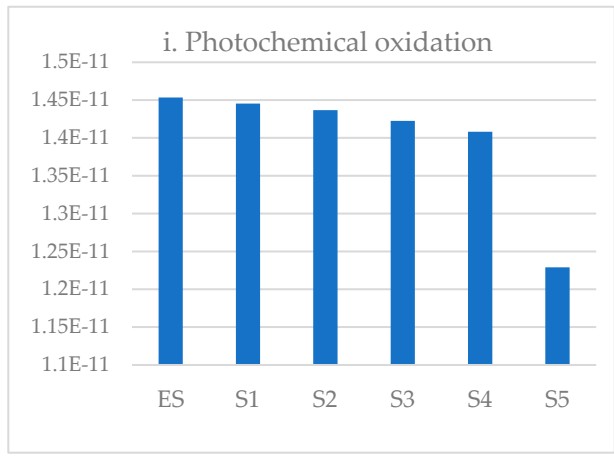

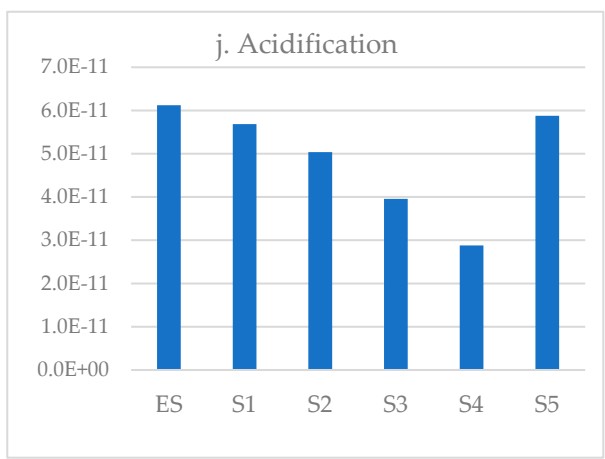

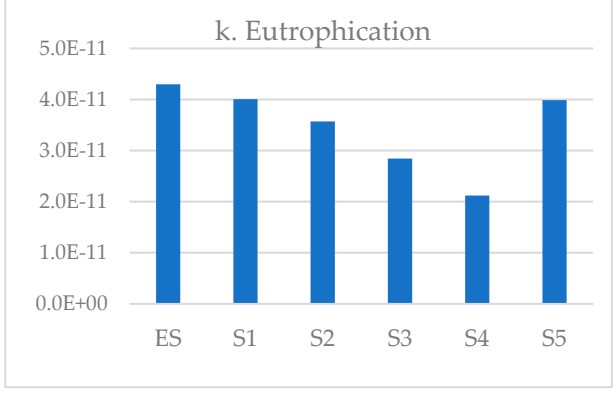

**Figure 8.** Comparative analysis of alternative scenarios and Existing System in terms of (**a**) abiotic depletion, (**b**) abiotic depletion (fossil), (**c**) global warming (GWP100a), (**d**) ozone layer depletion (ODP), (**e**) human toxicity, (**f**) freshwater aquatic ecotoxicity, (**g**) marine aquatic ecotoxicity, (**h**) terrestrial ecotoxicity, (**i**) photochemical oxidation, (**j**) acidification, (**k**) eutrophication.

**Table 4.** Impact assessment results of Existing System and alternative scenarios with CML-IA method.

| | Abiotic Depletion | Abiotic Depletion (Fossil Fuels) | Global Warming (GWP100a) | Ozone Layer Depletion (ODP) | Human Toxicity | Freshwater Aquatic Ecotoxicity | Marine Aquatic Ecotoxicity | Terrestrial Ecotoxicity | Photochemical Oxidation | Acidification | Eutrophication |
|---|---|---|---|---|---|---|---|---|---|---|---|
| ES | $8.31 \times 10^{-12}$ | $1.73 \times 10^{-10}$ | $6.91 \times 10^{-11}$ | $2.37 \times 10^{-13}$ | $1.71 \times 10^{-11}$ | $2.12 \times 10^{-10}$ | $2.85 \times 10^{-9}$ | $9.13 \times 10^{-11}$ | $1.45 \times 10^{-11}$ | $6.12 \times 10^{-11}$ | $4.3 \times 10^{-11}$ |
| S1 | $7.53 \times 10^{-12}$ | $1.65 \times 10^{-10}$ | $6.5 \times 10^{-11}$ | $2.1 \times 10^{-13}$ | $1.57 \times 10^{-11}$ | $1.97 \times 10^{-10}$ | $2.62 \times 10^{-9}$ | $9.04 \times 10^{-11}$ | $1.45 \times 10^{-11}$ | $5.68 \times 10^{-11}$ | $4.0 \times 10^{-11}$ |
| S2 | $6.37 \times 10^{-12}$ | $1.53 \times 10^{-10}$ | $5.9 \times 10^{-11}$ | $1.69 \times 10^{-13}$ | $1.37 \times 10^{-11}$ | $1.74 \times 10^{-10}$ | $2.27 \times 10^{-9}$ | $8.9 \times 10^{-11}$ | $1.44 \times 10^{-11}$ | $5.04 \times 10^{-11}$ | $3.57 \times 10^{-11}$ |
| S3 | $4.43 \times 10^{-12}$ | $1.33 \times 10^{-10}$ | $4.89 \times 10^{-11}$ | $1.01 \times 10^{-13}$ | $1.04 \times 10^{-11}$ | $1.37 \times 10^{-10}$ | $1.7 \times 10^{-9}$ | $8.67 \times 10^{-11}$ | $1.42 \times 10^{-11}$ | $3.96 \times 10^{-11}$ | $2.84 \times 10^{-11}$ |
| S4 | $2.48 \times 10^{-12}$ | $1.13 \times 10^{-10}$ | $3.88 \times 10^{-11}$ | $3.39 \times 10\text{-}14$ | $7.15 \times 10^{-12}$ | $9.98 \times 10^{-11}$ | $1.13 \times 10^{-9}$ | $8.44 \times 10^{-11}$ | $1.41 \times 10^{-11}$ | $2.88 \times 10^{-11}$ | $2.12 \times 10^{-11}$ |
| S5 | $1.0 \times 10^{-11}$ | $1.69 \times 10^{-10}$ | $6.68 \times 10^{-11}$ | $2.39 \times 10^{-13}$ | $1.66 \times 10^{-11}$ | $2.02 \times 10^{-10}$ | $2.72 \times 10^{-9}$ | $9.08 \times 10^{-11}$ | $1.23 \times 10^{-11}$ | $5.88 \times 10^{-11}$ | $3.99 \times 10^{-11}$ |

- Abiotic depletion

Abiotic depletion is a factor for evaluating the potential consequences of exhausting non-living resources like minerals. Upon analysis across scenarios, the greatest impact is noted in S5, where solar energy usage is presumed. Following closely is the Existing System, indicating significant depletion. Additionally, the utilization of recycled materials appears to alleviate the impact, as illustrated in Figure 8a.

- Abiotic depletion (fossil)

Abiotic depletion serves as a metric enabling the evaluation of the potential ramifications stemming from the depletion of non-living resources like fossil fuels. Analysis across scenarios reveals the highest figures within both the Existing System and S5, where solar energy adoption is assumed. However, given that this heightened impact is not evident during solar energy usage, it is theorized to originate from the production phase of

solar panels. Nonetheless, it is worth noting a decrease in impact with an increase in the utilization of recycled materials, as depicted in Figure 8b.

- Global warming (GWP100a)

Global warming potential (GWP) is a metric utilized to evaluate how much a substance contributes to global warming over a century-long period. It is calculated by assessing its capacity to retain heat in the atmosphere, enabling us to implement measures aimed at mitigating the environmental impact associated with climate change. In Scenario 5, where solar energy is presumed to be integrated into the current system, we witness a comparable effect on global warming over a 100-year span. Nonetheless, it is noteworthy that in scenarios assuming the utilization of recycled materials, there is a reduction in impact as depicted in Figure 8c as the proportion of recycled materials increases.

- Ozone layer depletion (ODP)

In the context of ozone layer depletion, we witness similar outcomes as those observed with global warming. In Scenario 5, where solar energy is presumed to be integrated into the Existing System, we observe parallel results. It is understood that the impact of solar energy usage on ozone layer depletion is minimal. We anticipate that this impact originates from the production process of solar panels. However, in scenarios involving the use of recycled materials, we observe a reduction in impact as the proportion of recycled materials increases, as illustrated in Figure 8d.

- Human toxicity

Human toxicity serves as a metric for evaluating the potential adverse effects of a product on human health over its entire life cycle. While the highest impact is observed in the Existing System, Scenario 5, where solar energy is assumed to be used, exhibits a similar effect to the ES. When compared with the results of PET/metallized PET/PE, we again attribute the difference to the disparity in material type. Additionally, the use of recycled materials significantly reduces the impact as seen in Figure 8e.

- Freshwater aquatic ecotoxicity

Freshwater aquatic ecotoxicity serves as a metric to evaluate the negative impacts on freshwater ecosystems. Upon scrutinizing the findings presented in Figure 8f, we discern divergent results from PET/metallized PET/PE observations. While we observe the highest impact in the Existing System, we also see a similar outcome in Scenario 5, where solar energy is assumed to be used. Additionally, it is clearly evident in Figure 8f that the impact decreases as the use of recycled materials increases. It is believed that there are two reasons for the difference in the results between the two packaging materials. Firstly, it is thought to be due to the differences in production processes, and secondly, it is attributed to the differences in material types.

- Marine aquatic ecotoxicity

Marine aquatic ecotoxicity is an impact factor that allows us to assess the adverse effects in marine water ecosystems. We observe the highest impact in the Existing System. In Scenario 5, where we assume the use of solar energy, we also observe results similar to the Existing System. When scenarios assuming the use of recycled materials are examined, we also see that the impact decreases as the amount of recycled material increases as given in Figure 8g.

- Terrestrial ecotoxicity

Terrestrial ecotoxicity is an impact factor that enables us to observe the adverse effects on terrestrial ecosystems. Upon examining Figure 8h, we notice that the highest impact is observed in the Existing System. Additionally, as expected, it is observed that the impact decreases as the use of recycled materials increases. In Scenario 5, where solar energy is used, a slight decrease in impact compared to the Existing System is also observed.

- Photochemical oxidation

Photochemical oxidation is a type of chemical reaction where chemical compounds undergo oxidation or breakdown using light energy. These reactions are triggered by sunlight or other light sources and play a significant role in the formation of air pollution, the atmospheric chemical transformation of organic compounds, and various industrial applications. It is an impact factor that emerges as a component of air pollution. Upon examining this impact factor, it has been observed that the highest impact occurs in the Existing System. Additionally, it has been noticed that the impact decreases slightly when recycled materials are used. Particularly, a very significant decrease has been observed in the assumed scenario where solar energy is used, as seen in Figure 8i.

- Acidification

Acidification is an impact factor that describes the phenomenon of elevating the environmental acidity of a substance. It encompasses all environmental media including soil, water, and air. We observe the highest impact in both the Existing System and in Scenario 5, where solar energy is assumed to be utilized. Moreover, we hypothesize that the impact in the solar energy system arises not during its operation but rather during the production of solar panels. In scenarios employing recycled materials, it has been noted that as the utilization of recycled materials rises, the impact decreases, as illustrated in Figure 8j.

- Eutrophication

Eutrophication is an impact factor that allows us to witness the negative consequences stemming from an overabundance of nutrients in aquatic ecosystems. The greatest impact is noted within the Existing System. Nonetheless, the utilization of recycled materials markedly diminishes this impact, as depicted in Figure 8k. Additionally, solar energy exhibits a high impact similar to Scenario 1, which employed 10% recycled raw materials.

## 4. Discussion

When cumulative energy demand analysis was conducted for PET/metallized PET/PE packaging, the expected outcomes were obtained for both ES and the five scenarios. These results provide findings consistent with Türkiye's energy profile, where the majority of energy needs are met by fossil fuels. Among the scenarios, the lowest energy demand for one lot of packaging was observed in Scenario 4, where 75% of the input material was assumed to be recycled and the remaining 25% was virgin material. Based on these results, it emerged that recycling plastics used as raw materials, reducing waste, and incorporating them into the process are a more suitable solution than converting the energy source to a renewable one.

When 11 impact factors analyzable by the CML-IA method were examined for PET/metallized PET/PE packaging, no negative impacts were observed, yet the highest impact was seen in the "marine aquatic ecotoxicity" factor for all scenarios and the ES. This effect was highest in the ES and gradually increased in scenarios with increased recycling rates. This effect was followed by "abiotic depletion (fossil fuels)" in second place. Although changes among the scenarios were minor, the least impact was observed in Scenario 4. The third-ranked "freshwater aquatic ecotoxicity" factor and the fourth-ranked "global warming" factor also showed the least impact in Scenario 4. Considering all other impact factors, it was observed that using recycled materials had a greater impact reduction compared to using renewable solar energy. Similar to the findings of the CED analysis, it was concluded that the most suitable option for this type of packaging is to increase the use of recycled materials instead of using renewable energy sources.

When analyzed using the cumulative energy demand method, PP/metallized PP/PP packaging yields different results compared to PET/metallized PET/PE packaging. Although the need for non-renewable fossil energy remains highest, we observe greater variations in scenarios for this packaging type. It is believed that this difference stems from the types of plastics used as raw materials. The Existing System (ES) has the highest

demand for fossil energy, followed by Scenario 5, assuming the use of solar energy. Despite the least impact observed in the scenario where 75% of the input mass is assumed to be recycled material, it has been observed that the impact decreases as the proportion of recycled material increases. While similar results are obtained for other energy types, a single-score analysis reveals that there is not a significant change in the total energy demand when solar energy is used. Instead, more significant changes can be achieved through increased use of recycled materials, making Scenario 4 the most optimal option for this packaging type.

When analyzed with the CML-IA method, PP/metallized PP/PP packaging shows the highest values in the "marine aquatic ecotoxicity" factor, similar to PET/metallized PET/PE packaging. Changes are more pronounced here as well. It is observed that the impact decreases as the recycling rate increases, and in scenarios where solar energy is used, the impact is close to the highest value in the ES. Additionally, it is noteworthy that Scenario 4, assuming 75% recycled material, exhibits the least impact. The second highest factor is "freshwater aquatic ecotoxicity", followed by "abiotic depletion (fossil fuels)". It can be said that Scenario 4 has the least impact on these factors.

Comparing the results of the two packaging types, it is clear that the differences stem from the types of plastics used as raw materials. However, it can be stated, according to both analysis methods, that increasing the proportion of recycled material is the best option for both packaging types. While using renewable solar energy may not yield significantly better results than the current production method, it should be noted that the impact obtained from recycling is much greater.

## 5. Conclusions

This study holds significant academic merit primarily due to its reliance on actual data. Moreover, its contribution to the existing literature is noteworthy as it offers a comprehensive overview by juxtaposing two distinct flexible packaging alternatives and evaluating them through diverse methodological lenses. Originating from a doctoral dissertation, the study endeavors to furnish a holistic understanding by endorsing life cycle costing (LCC) and social life cycle assessment (SLCA).

The principal findings of this investigation can be summarized as follows:

The comparative analysis revealed that the environmental performance of the prevailing system failed to surpass the efficacy of the other five scenarios across both flexible packaging variants. Upon scrutinizing the outcomes derived from each methodological approach, Scenario 4, characterized by used 75% recycled raw material, emerged as the most optimal sustainable solution.

Differential impacts observed between the two packaging types predominantly stemmed from the variation in plastic composition. This observation underscores the potential efficacy of transitioning towards less environmentally impactful plastics within flexible packaging, signifying a potential avenue for mitigating environmental burdens.

Evaluation of S5, wherein solar energy was integrated into both packaging types, elucidated that despite its renewable nature, solar energy failed to mitigate environmental impacts within the impact factor categories as per the CED and CML-IA methodologies. Consequently, the findings advocate for a reevaluation of strategies, positing that prioritizing recycling within current production paradigms may yield more substantial environmental benefits than solely relying on renewable energy sources. Furthermore, exploring the synergistic effects of augmenting recycling rates alongside renewable energy sources warrants consideration as a prospective research avenue.

Noteworthy is the revelation from the CML-IA analysis that all impacts were positive for both packaging options, underscoring the imperative for proactive interventions to mitigate environmental repercussions. Strategies such as diversifying raw material types, transitioning to renewable energy sources, and minimizing waste through the incorporation of recycled materials are deliberated within this discourse. Moreover, the proposition of a composite scenario integrating these variables presents a compelling proposition for future inquiry.

While this study represents a pivotal contribution to extant literature by virtue of its empirical grounding, a comprehensive elucidation can be attained through further empirical investigations employing LCC and SLCA methodologies.

**Author Contributions:** Conceptualization, B.T.-Ç., G.A.Ç. and E.Y.-G.; methodology, B.T.-Ç. and E.Y.-G.; analysis, B.T.-Ç.; data curation, B.T.-Ç. and G.A.Ç.; writing—original draft, B.T.-Ç.; writing—review and editing, B.T.-Ç.; visualization, B.T.-Ç.; supervision, G.A.Ç. All authors have read and agreed to the published version of the manuscript.

**Funding:** This research received no external funding.

**Institutional Review Board Statement:** Not applicable.

**Informed Consent Statement:** Not applicable.

**Data Availability Statement:** Data are contained within the article.

**Acknowledgments:** This study was conducted in collaboration with Marmara University and the R&D Center of Elif Plastik Ambalaj Sanayi ve Tic. A.Ş.-Huhtamaki Flexibles Istanbul.

**Conflicts of Interest:** The authors declare no conflicts of interest.

## Appendix A

**Table A1.** Production Data of PET/Metallized PET/PE.

| Process 1: Blown Film Extrusion | |
|---|---|
| Electric Consumption Trigen (kwh) | 757.27 |
| Electric Consumption Grid (kwh) | 0.42 |
| Pressured Air (m$^3$) | 43.20 |
| Chiller (kwh) | 292.57 |
| Scrap (kg) | - |
| Input Raw Material | |
| Grade Name | Consumption (kg) |
| PE Type 1 | 191.15 |
| PE Type 2 | 99.13 |
| PE Type 3 | 34.13 |
| Masterbatch | 3.28 |
| Process 2: Flexo Printing | |
| Electric Consumption Trigen (kwh) | 88.54 |
| Electric Consumption Grid (kwh) | 9.91 |
| Pressured Air (m$^3$) | 11.48 |
| Heat Energy (kwh) | 172.98 |
| Ink Consumption (kg) | 4.97 |
| Solid ink. kg | 2.27 |
| Solvent Consumption (kg) | 2.70 |
| Scrap (kg) | - |
| Input Raw Material | |
| Grade Name | Consumption (kg) |
| PET Film | 122.72 |
| Process 3: Lamination 1 | |
| Electric Consumption Trigen (kwh) | 19.07 |
| Electric Consumption Grid (kwh) | 8.16 |
| Pressured Air (m$^3$) | 5.86 |
| Heat Energy (kwh) | 204.96 |
| Chiller (kwh) | 7.55 |

**Table A1.** *Cont.*

| Input Raw Material | |
|---|---|
| Grade Name | Consumption (kg) |
| Metallized PET Film | 81.09 |
| Solvent | 9.12 |
| Lamination Adhesive Type 1 | 14.15 |
| Lamination Adhesive Type 2 | 7.07 |
| Process 4: Lamination 2 | |
| Electric Consumption Trigen (kwh) | 16.21 |
| Electric Consumption Grid (kwh) | 6.94 |
| Pressured Air (m$^3$) | 3.60 |
| Chiller (kwh) | 4.64 |
| Scrap (kg) | - |
| Input Adhesive Material | |
| Grade Name | Consumption (kg) |
| Lamination Adhesive Type 3 | 8.83 |
| Lamination Adhesive Type 4 | 3.98 |
| Process 5: Slitting/Dilimleme | |
| Electric Consumption Trigen (kwh) | 7.92 |
| Electric Consumption Grid (kwh) | 3.37 |
| Pressured Air (m$^3$) | 7.43 |
| Scrap (kg) | - |
| Process 6: Converting | |
| Electric Consumption Trigen (kwh) | 83.64 |
| Electric Consumption Grid (kwh) | 31.75 |
| Pressured Air (m$^3$) | 1443.42 |
| Chiller (kwh) | 7.29 |
| Scrap (kg) | - |
| Input Raw Material | |
| Name | Consumption (kg) |
| Zipper | 18.56 |

**Table A2.** Production Data of PP/Metallized PP/PP.

| Process 1: Flexo Printing | |
|---|---|
| Electric Consumption Trigen (kwh) | 12.70 |
| Electric Consumption Grid (kwh) | 1.43 |
| Pressured Air (m$^3$) | 3.27 |
| Heat Energy (kwh) | 48.97 |
| Ink Consumption (kg) | 7.75 |
| Solvent Consumption (kg) | 4.21 |
| Scrap (kg) | - |
| Input Raw Material | |
| Grade Name | Consumption (kg) |
| PP Film Type 1 | 82.47 |
| Process 2: Lamination 1 | |
| Electric Consumption Trigen (kwh) | 8.46 |
| Electric Consumption Grid (kwh) | 3.62 |
| Pressured Air (m$^3$) | 1.92 |
| Chiller (kwh) | 2.48 |
| Scrap (kg) | - |

**Table A2.** *Cont.*

| Input Raw Material | |
|---|---|
| Grade Name | Consumption (kg) |
| Metallized PP Film | 33.96 |
| Lamination Adhesive Type 5 | 7.1 |

| Process 3: Lamination 2 | |
|---|---|
| Electric Consumption Trigen (kwh) | 8.45 |
| Electric Consumption Grid (kwh) | 3.62 |
| Pressured Air (m$^3$) | 1.84 |
| Chiller (kwh) | 2.37 |
| Scrap (kg) | - |

| Input Raw Material | |
|---|---|
| Grade Name | Consumption (kg) |
| PP Film Type 2 | 114.03 |
| Lamination Adhesive Type 6 | 7.100 |

| Process 4: Slitting | |
|---|---|
| Electric Consumption Trigen (kwh) | 1.50 |
| Electric Consumption Grid (kwh) | 0.63 |
| Pressured Air (m$^3$) | 3.31 |
| Scrap (kg) | - |

| Process 5: Converting | |
|---|---|
| Electric Consumption Trigen (kwh) | 39.66 |
| Electric Consumption Grid (kwh) | 14.61 |
| Pressured Air (m$^3$) | 772.20 |
| Chiller (kwh) | 3.9 |
| Scrap (kg) | - |

| Input Raw Material | |
|---|---|
| Name | Consumption (kg) |
| Zipper | 8.40 |

**Table A3.** Recycled Raw Material Input Amounts for PET/Metallized PET/PE.

| PET/Metallized PET/PE | | | | | | |
|---|---|---|---|---|---|---|
| Process | ES | S1 | S2 | S3 | S4 | S5 |
| Extrusion | 191.15 kg PE Type 1 99.13 kg PE Type 2 | 172 kg PE Type 1 89.2 kg PE Type 2 19.1 kg recycled PE Type 1 9.91 kg recycled PE Type 2 | 143 kg PE Type 1 74.3 kg PE Type 2 47.8 kg recycled PE Type 1 24.8 kg recycled PE Type 2 | 95.6 kg PE Type 1 49.5 kg PE Type 2 95.6 kg recycled PE Type 1 49.6 kg recycled PE Type 2 | 48.2 kg PE Type 1 24.8 kg PE Type 2 74.3 kg recycled PE Type 1 143 kg recycled PE Type 2 | Same as ES |
| Printing | 122.72 kg PET | 110 kg PET 12.3 recycled PET | 92 kg PET 30.7 recycled PET | 61.3 kg PET 61.4 recycled PET | 30.7 kg PET 92 recycled PET | Same as ES |
| Lamination 1 | - | - | - | - | - | - |
| Lamination 2 | - | - | - | - | - | - |
| Slitting | - | - | - | - | - | - |
| Converting | - | - | - | - | - | - |

**Table A4.** Recycled Raw Material Input Amounts for PP/Metallized PP/PP.

| PP/Metallized PP/PP | | | | | | |
|---|---|---|---|---|---|---|
| Process | ES | S1 | S2 | S3 | S4 | S5 |
| Printing | 82.47 kg PP Type 1 | 74.2 kg PP Type 1 8.25 recycled PP Type 1 | 61.9 kg PP Type 1 20.6 recycled PP Type 1 | 41.3 kg PP Type 1 41.2 recycled PP Type 1 | 20.6 kg PP Type 1 61.9 recycled PP Type 1 | Same as ES |
| Lamination 1 | - | - | - | - | - | - |
| Lamination 2 | 114.030 kg PP Type 2 | 103 kg PP Type 2 11.4 kg recycled PP Type 2 | 85.5 kg PP Type 2 28.5 kg recycled PP Type 2 | 57 kg PP Type 2 257 kg recycled PP Type 2 | 28.5 kg PP Type 2 85.5 kg recycled PP Type 2 | Same as ES |
| Slitting | - | - | - | - | - | - |
| Converting | - | - | - | - | - | - |

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
