# Peer review of "Environmental Life Cycle Assessment of Two Types of Flexible Plastic Packaging under a Sustainable Circular Economy Approach"

_sustainability, doi:10.3390/su16083149_

Round 1

Reviewer 1 Report

Comments and Suggestions for Authors

The research in the manuscript entitled "Environmental Life Cycle Assessment of Two Types of Flexible Plastic Packaging Under a Circular Economy Approach" assesses and compares the environmental impact of two types of flexible plastic packaging. The aim was to find the optimal production scenario with the least environmental impact using the life cycle assessment methodology. Various scenarios were studied assuming that changes would be made in the production stages of two flexible packages. The obtained results showed the largest environmental impact was in the analyzed „Existing System (ES)” (based on actual production data from Huhtamaki Flexibles, Turkey). The scenario with solar energy (S5) had almost as much impact as the ES.

In the abstract, only the main conclusions should be indicated. I suggest changing the abstract in this respect.

I suggest changing all keywords to ones other than those in the title.

The choice of the Life Cycle Assessment (LCA) method has been appropriate. This method enables in a modern way the evaluation and comparison of the different technology methods and assesses their impact on the environment throughout the entire life cycle of the products. The description of the methodology is detailed. However, the Authors did not provide any information about the ecoinvent.

The results are presented understandably. However, the quality of the presentation is not high

A broader literature review would allow readers to understand better the context of research and its relation to other relevant studies. Additionally, it will demonstrate a deeper engagement with the existing body of knowledge, making this work more robust and relevant.

The conclusions are consistent with the evidence and arguments presented.

The use of an additional complementary methodology such as the Life Cycle Costing could have provided a more comprehensive understanding of the sustainability of two types of flexible plastic packaging by integrating environmental and economic aspects in the analysis. This could provide perspectives beyond the environmental impact by considering other important dimensions related to circularity.

Certain parts of the text should be improved.

Line 20-23: It should be changed, without „ES”, „S4”, and „S5”.

Line 36: „Türkiye” should be changed to „Turkey”.

Line 43, 84, 110: The abbreviations should be explained when they first appear in the text, e.g. polypropylene (PP), life cycle assessment (LCA), etc. After that, only the abbreviations are used in the text.

Line 75: Missing information about the source of Figure 1.

Line 88, 298: The dot should be at the end of the sentence.

Lines 127-298: I suggest removing the division into subsections.

Line 380, 540, 549: The figures are not legible.

Lines 404-490: I suggest removing the division into subsections.

Line 453: „Marina” should be changed to „Marine”.

Lines 575-688: I suggest removing the division into subsections.

Lines 692-740: There are no references to literature in the discussion.

Lines 749-776: I suggest removing the bullet format.

Lines 786-829: The list of references should be prepared according to the journal's requirements.

Author Response

Dear Reviewer,

You can find our answers to your valuable comments in the attachment.

Best regards,

Reviewer 2 Report

Comments and Suggestions for Authors

In this manuscript, authors conducted a LCA on the production process of two flexible packaging structures. The LCA focuses on how scrap recycling or implement of solar energy affects the CED and CML-IA.

The manuscript is written in well-structured and written in smooth language. However, some critical factors (see comments below) are missing and thus lead to a strongly biased conclusion. I do not suggest publishing this manuscript in current form.

11. Authors did not define scrap rate, which is scrap rate% = scrap weight/total produced weight*100. If the scrap rate of production is as low as 0.1%, no matter how much is recycled, there will be little impact. The “60%PE and 40% PET” provided in the manuscript is not called scrap rate, instead, it is just scrap composition/content.

22. Authors did not define how recycling is done. The scraps have ink, adhesive and even bonded structure. There is no way to re-use it directly. There must be some cleaning, separation, sorting, drying, melting and reprocessing to be able to use again. Is it even separable once PE and PET are bonded together? All these contribute significant amount of CED and CML-IA consideration. If all these are considered, author may even have an opposite conclusion.

33. Minor comment: the use of decimal point is not consistent. Sometimes coma was used while sometimes point was used. For example, line 58 on page 2, Figure 6-9 and Table 2-3.

Author Response

Dear Reviewer,

Attached please find our answers to your valueable comments.

Best regards,

Reviewer 3 Report

Comments and Suggestions for Authors

There are fundamental comments, there are comments on the formatting of the article.

The authors use two different packaging. At the same time, they encrypt them. The process of decomposition of a polymer material is continuously associated with a change in the structure of the polymer. A question arises about the decomposition mechanism. At the same time, it is not clear how an expert should evaluate this article. There are hundreds of papers in which the authors use various methods to understand the degradation processes of polymers, such as IR spectroscopy, DSC calorimetry and many others. This article doesn't have that at all. Moreover, the authors did not even say what kind of material they were analyzingThe authors give a general class of materials, for example PET, without specifying the brand. In connection with this, the article is very difficult to evaluate.

Authors should also carefully read the instructions for authors to properly format the article in accordance with the requirements of the journal, namely:

1.      Titles for figures must be in the same format according to the instructions for authors.

2.      The title of Table 3 should appear before the table, not after it.

3.      Make changes to the annotations, namely clarify abbreviations «ES», «S4», «S5»

4.      Eliminate repetition of keywords: «LCA, life cycle assessment (LCA

5.      In the text, reference numbers should be placed in square brackets [ ], and placed before the punctuation; for example [1], [1–3] or [1,3].

6.      Prepare a list of references according to the instructions for authors

Author Response

Dear Reviewer,

Thank you for your valuable comments to our manuscript. Attached please find our answers.

Best regards,

Reviewer 4 Report

Comments and Suggestions for Authors

In order to protect the environment. in this manuscript, the author performed a life cycle assessment of two flexible plastic packaging, The purpose of the study is to find the best production option with the least environmental impact.

1. When authors focused on the discussion of LCA of two types of flexible packing within a circular economy, the discussion lacked detailed description and comparison with each other, especially greenhouse gas emissions and water consumption.

2. For both metallized PET and metallized CPP, the authors considered the aluminum content to be nanoscale, so the aluminum content was ignored. This may not be appropriate.

3. How are the recovery rates calculated in the various recycling schemes?

4. The analysis in part 3.1.1.3 was divided into several short parts, which are difficult to view and compare for readers.

5. Some of the date in Table 2 is the same, but not the same in the correspomding graph.

6. Where is the actual production data for 1 lot from Huhtamaki Flexibles Türkiye used by ES?

7. Several acronyms such as CML-IA, ES, OPP, and CPP appear in the introduction to this manuscript without giving their full names. This can create a reading barrier for readers who are not very familiar with the field.

Author Response

Dear Reviewer,

Thank you very much for your valuable comments. You can find all our responses attached.

Best regards,

Round 2

Reviewer 1 Report

Comments and Suggestions for Authors

Dear Authors,

The manuscript has been sufficiently improved.

Author Response

Dear Reviewer,

Thank you for your valuable comments and support for the improvement of our article.

Reviewer 2 Report

Comments and Suggestions for Authors

Unfortunately, authors failed to address my two major comments from the previous report. Both of them still hold true after the revision.

(1) The author is still misusing the term "scrap rate". Scrap rate = (weight of the scrap) / (weight of total feeding raw material). I am disappointed to see that authors failed to understand my previous comment and failed to calculate scrap rates properly. 

(2) This paper keeps highlighting that the analysis is based on a real-world industrial case, which is completely misleading. The whole "recycling" imagined by authors doesn't exist yet in the real world. There is no easy way to recycle any scrapes with all the printed ink and cured adhesives. A huge amount of energy and waste water from recycling is ignored which is a critical failure in LCA. 

Authors claimed that they are adding such calculations to their new paper. However, that doesn't change the fact that the current manuscript is fatally flawed and unpublishable.

In conclusion, this paper has strongly biased conclusion due to a wrongful assumption that scraps can be recycled. This paper also doesn't contribute any value to either academia or industry. For academia, the analysis model is too basic and there are critical flaws in LCA model. For industry, the report completely ignored the scrap rate and recycling feasibility.

Comments on the Quality of English Language

None.

Author Response

Dear Reviewer,

Best regards,

Börçe TUNÇOK ÇEÅžME

Reviewer 3 Report

Comments and Suggestions for Authors

The authors have made the necessary corrections and clarifications to the article. I recommend this article for publication

Author Response

Dear Reviewer,

Thank you for taking the time to contribute to the development of our article and for your support.

Best regards,

Börçe TUNÇOK ÇEÅžME

Round 3

Reviewer 2 Report

Comments and Suggestions for Authors

Authors managed to address the two concerns by changing the study subject from recycling product to using recycled materials. I only have one comment for this manuscript as below. 

What are the raw input numbers? Authors claimed to use a set of real industry data but it is crucial to list them out, so readers can understand where the results come from. It can be in a supplementary material but cannot be omitted.

Author Response

Dear Reviewer,

Best regards,

Börçe TUNÇOK-ÇEÅžME
